# Programmatically Grounded, Compositionally Generalizable Robotic Manipulation

**Renhao Wang**[1*]  **Jiayuan Mao**[2*]  **Joy Hsu**[3]  **Hang Zhao**[1,4,5]  **Jiajun Wu**[3]  **Yang Gao**[1,4,5]

[1]Tsinghua University  [2]MIT  [3]Stanford University
[4] Shanghai Artificial Intelligence Laboratory  [5] Shanghai Qi Zhi Institute

## Abstract

Robots operating in the real world require both rich manipulation skills as well as the ability to semantically reason about when to apply those skills. Towards this goal, recent works have integrated semantic representations from large-scale pretrained vision-language (VL) models into manipulation models, imparting them with more general reasoning capabilities. However, we show that the conventional *pretraining-finetuning* pipeline for integrating such representations entangles the learning of domain-specific action information and domain-general visual information, leading to less data-efficient training and poor generalization to unseen objects and tasks. To this end, we propose PROGRAMPORT, a *modular* approach to better leverage pretrained VL models by exploiting the syntactic and semantic structures of language instructions. Our framework uses a semantic parser to recover an executable program, composed of functional modules grounded on vision and action across different modalities. Each functional module is realized as a combination of deterministic computation and learnable neural networks. Program execution produces parameters to general manipulation primitives for a robotic end-effector. The entire modular network can be trained with end-to-end imitation learning objectives. Experiments show that our model successfully disentangles action and perception, translating to improved zero-shot and compositional generalization in a variety of manipulation behaviors. Project webpage at: https://progport.github.io.

## 1 Introduction

Robotic manipulation models that map directly from raw pixels to actions are capable of learning diverse and complex behaviors through imitation. To enable more abstract goal specification, many such models also take as input natural language instructions. However, this *vision-language manipulation* setting introduces a new problem: the agent must jointly learn to ground language tokens to its perceptual inputs, and correspond this grounded understanding with the desired actions. Moreover, to fully leverage the flexibility of language, the agent must handle novel vocabulary and compositions not explicitly seen during training, but specified at test time (Fig. 1).

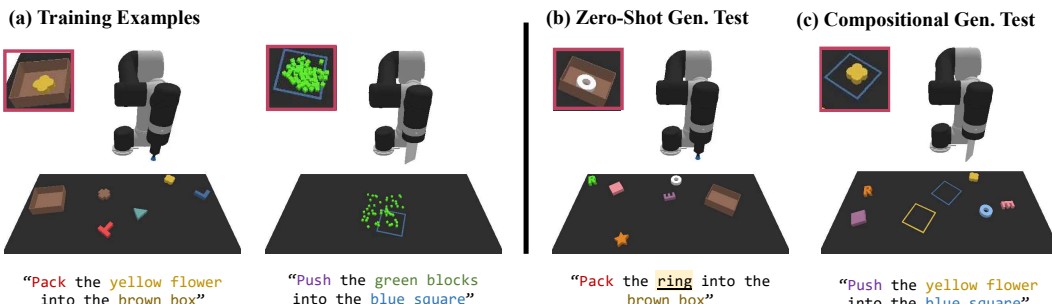

Figure 1: **Zero-Shot and Compositional Generalization:** Our framework, PROGRAMPORT, is capable of generalizing to *combinations* of *unseen* objects and manipulation behaviors at test time.

---

[*]equal contribution

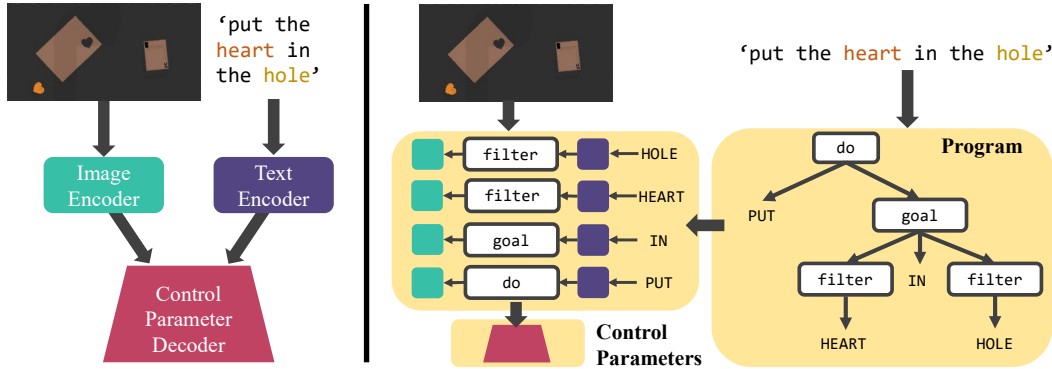

**(a) Pretraining-Finetuning Paradigm**   **(b) Program-based Modular Paradigm (ProgramPort)**

Figure 2: **(a)** Existing manipulation models entangle visual grounding from pretrained VL models with task-specific policy learning. **(b)** Our disentangled approach leverages program-based representation of language semantics to yield faithful VL-grounding and improved genearlization.

To these ends, many recent works have relied on large pretrained vision-language (VL) models such as CLIP (Radford et al., 2021) to tackle both grounding and zero-shot generalization. As shown in Fig. 2a, these works generally treat the pretrained VL model as a semantic prior, for example, by partially initializing the weights of image and text encoders with a pretrained VL model (Ahn et al., 2022; Khandelwal et al., 2022; Shridhar et al., 2022a). These models are then updated via imitating expert demonstrations. However, this training scheme entangles the learning of domain-specific control policies and domain-independent vision-language grounding. Specifically, we find that these VL-enabled agents overfit to their task-specific data, leveraging shortcuts during training to successfully optimize their imitation learning (IL) objective, without learning a generalizable grounding of language to vision. This phenomenon is particularly apparent when the agent is given language goals with unknown concepts and objects, or novel compositions of known concepts and objects. For example, an agent that overfits to packing shapes into a box can fail to generalize to other manipulation behaviors (e.g. pushing) involving the same shapes.

In this work, we introduce PROGRAMPORT, a program-based modular approach that enables more faithful vision-language grounding when incorporating pretrained VL models for robotic manipulation. Given the natural language instruction, we first use a Combinatory Categorial Grammar (CCG) (Steedman, 1996) to parse the sentence into a "manipulation program," based on a compact but general domain-specific language (DSL). The program consists of functional modules that are either visual grounding modules (e.g., locate all objects of a given category) or action policies (e.g., produce a control parameter). This enables us to directly leverage a pretrained VL model to ground singular, independent categories or attribute descriptors to their corresponding pixels, and thus disentangles the learning of visual grounding and action policies (Fig. 2b).

Our programmatically structured, modular design enables PROGRAMPORT to successfully learn more performant imitation policies with fewer data across 10 diverse tasks in a tabletop manipulation environment. We show that after training on a subset of objects, visual properties, and actions, our model can zero-shot generalize to completely different subsets and reason over novel compositions of language descriptors, without further finetuning (Fig. 1).

## 2 BACKGROUND

**Problem formulation**. We take a manipulation primitive-based approach to robot learning. At a high level, we assume our robot is given a set of *primitives* $\mathcal{P}$ (e.g., pick, place, and push). Such primitives are usually defined by interaction modes between the robot and objects, parameterized by continuous parameters, and their composition spans a wide range of tasks. Throughout the paper, we will be using the simple primitive set $\mathcal{P} = \{pick, place\}$ in a tabletop environment as the example, and extend our framework to other primitives such as pushing in the experiment section.

Therefore, our goal is to learn a policy $\pi$ mapping an input observation $\mathbf{o}_t$ at time $t$ to an action $\mathbf{a}_t$. Each $\mathbf{a}_t$ is a tuple of two control parameters $(\mathcal{T}_{pick}, \mathcal{T}_{place})$, which, in the pick-and-place setting,

parameterizes the poses for the robot end effector when picking and placing objects, respectively. This formulation also generalizes to other primitives such as pushing, in which case the control parameters are the pre and post-push locations. More specifically, in a language-guided tabletop manipulation task, our observation is given by $\mathbf{o}_t = (\mathbf{x}_t, \mathbf{l}_t)$. Here, $\mathbf{x}_t \in \mathbb{R}^{H \times W \times 4}$ is the top-down orthographic RGB-D projection of the 3D visual scene, while $\mathbf{l}_t$ is a natural language description of the desired goal. The control parameters $\mathcal{T}_{pick}, \mathcal{T}_{place}$ both lie in $SE(2)$.

To learn such a policy, we assume access to a set of expert demonstrations $\mathcal{D} = \{d_1, d_2, \ldots, d_n\}$, each consisting of discrete time-indexed scene-language-action tuples so that $d_i = \{(\mathbf{x}_1, \mathbf{l}_1, \mathbf{a}_1), (\mathbf{x}_2, \mathbf{l}_2, \mathbf{a}_2), \ldots, (\mathbf{x}_j, \mathbf{l}_j, \mathbf{a}_j)\}$. The policy $\pi$ can be trained with an imitation learning objective by randomly sampling a particular tuple at time $t$ from any demonstration.

**Vision-language manipulation**. Akin to Transporter Networks (Zeng et al., 2020), our policy $\pi$ is composed of two functions, $\mathcal{Q}_{pick}$ and $\mathcal{Q}_{place}$, outputting dense pixelwise action values over $\mathbb{R}^{H \times W}$, and operating in sequence. $\mathcal{Q}_{pick}$ first decides a particular pixel location $\mathcal{T}_{pick}$ in the top-down visual field to pick the target object; $\mathcal{Q}_{place}$ then conditions on the pick location to decide a particular place pose $\mathcal{T}_{place}$ for the object:

$$\mathcal{T}_{pick} = \underset{(u,v) \in H \times W}{\arg\max} \, \mathcal{Q}_{pick}\left((u,v) \mid \mathbf{x}_t, \mathbf{l}_t\right), \quad \mathcal{T}_{place} = \underset{\tau_i \in \{\tau\}}{\arg\max} \, \mathcal{Q}_{place}\left(\tau_i \mid \mathbf{x}_t, \mathbf{l}_t, \mathcal{T}_{pick}\right) \quad (1)$$

where poses representing the manipulation end effector primitives are apriori discretized into the set $\{\tau\}$, composed of discrete pixel locations and a discrete set of 2D rotation angles. Recently, CLIPort (Shridhar et al., 2022a) proposed to integrate pretrained vision-language (VL) models into such frameworks. Shown in Fig. 2a, in CLIPort, both $Q_{pick}$ and $Q_{place}$ are based on two-stream fully-connected network (FCN) architectures. The encoders of the FCNs are initialized from pretrained weights of CLIP (Radford et al., 2021), while the control parameter decoders are randomly initialized and optimized. Similar paradigms have also been applied to other manipulation environments such as Khandelwal et al. (2022) and Ahn et al. (2022).

While the *pretraining-finetuning* framework can leverage features from the pretrained CLIP encoders to inform the model on a wide array of vision-language semantics, its monolithic structure entangles the learning of vision-language grounding and action policies. This has two drawbacks: poor data efficiency and overfitting. More specifically, since training data will only contain a finite number of compositions of action, category, and property terms, the model usually fails to generalize to manipulation instructions that contain novel compositions of seen terms and unseen terms.

## 3 PROGRAMPORT

The key idea of our framework PROGRAMPORT is that reliably utilizing pretrained vision-language (VL) models in manipulation involves leveraging the structure of natural language. Given the visual input and natural language instruction, PROGRAMPORT parses the instruction into a program based on a domain-specific language (DSL). Each generated program corresponds to a hierarchical structure of functional modules implemented by neural networks, each fulfilling a specific operation over the vision or action representation. The program executor executes the program upon the visual input and derives the control parameters to action primitives. This compositional design disentangles the learning for visual grounding modules (e.g., locating the objects being referred to) and the action modules (e.g., predicting the control parameters). Concretely, it enables us to directly leverage pretrained VL models such as CLIP as the visual grounding module without further finetuning, and to only focus on learning action modules, which improves the transparency, data efficiency, and the generalization of the system. A visual overview of our framework is available in Fig. 3.

In the remainder of this section, we first describe the structure of our manipulation programs, and how they are derived from natural language instructions (Section 3.1). We then detail the neural modules which realize their execution (Section 3.2). Finally, we specify the training paradigm (Section 3.2).

### 3.1 PROGRAMMATIC REPRESENTATIONS OF MANIPULATION INSTRUCTIONS

At the core of the proposed PROGRAMPORT framework is a programmatic representation for instructions. Illustrated in Fig. 2b, we use a program to represent the underlying semantics of the natural language instruction. Each program has a hierarchical structure composed of functional modules. Each functional module takes the outputs from its predecessors, the visual input, and optionally additional concepts from the language (e.g., *red*) and computes the outputs (e.g., an attention mask).

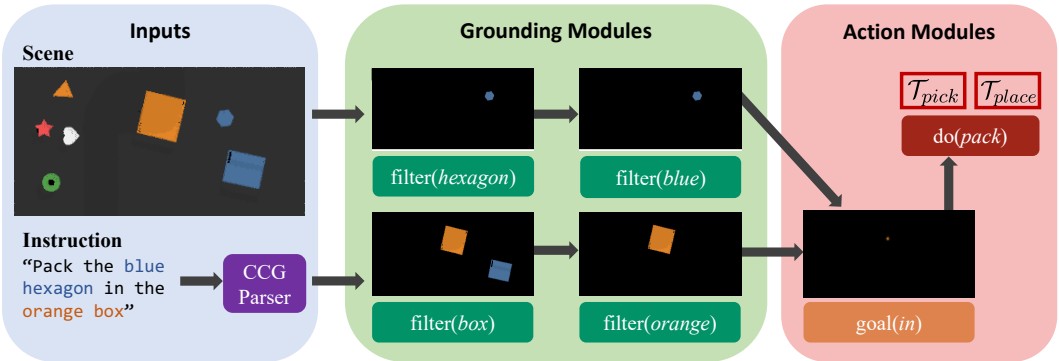

Figure 3: **The execution of a manipulation program.** A CCG parser first generates a manipulation program. The task-agnostic visual grounding modules then identify visual semantics associated with the desired objects and targets. Finally, the task-specific action modules leverage these grounding results to output the manipulation control parameters.

We present the detailed definition of our domain-specific language (DSL) in Appendix A.1. All functional modules can be categorized into two groups: *visual grounding modules* and *action modules*. *Visual grounding modules* take as input language descriptors, and return spatial masks over the corresponding locations in the visual scene. In this paper, the only visual grounding module we consider is the *filter* operation, which localizes objects that have certain properties (as in *filter(blue, ...)* will return blue objects in the scene). In contrast, *action modules* generate control parameters for robot primitives. They take as input the outputs from visual grounding modules. For example, the *goal* module generates a 2D pose that has a specified spatial relationship with another object (as in *goal(in, ...)* will return a location in the stated object). The *do(pack, ..., ...)* operator takes three arguments: the action name *pack*, the object to be packed, and the goal location, both specified as masks over the input image, and generates the control parameter for the robot's next move.

**Semantic parsing with a combinatory categorial grammar**. To parse language instructions into a program structure, we use the Combinatory Categorial Grammar (CCG) formalism (Steedman, 1996). A CCG parser is composed of a small (less than 10) set of universal categorial grammar rules and a domain-specific lexicon. Each lexicon entry maps a natural language word (e.g., *red*) to its syntactic type (e.g., *N/N*) and semantic form (e.g., $\lambda x.filter(red, x)$). The notation *N/N* indicates that this word can be combined with a word on its right with type *N* (noun), and the semantic form indicates that this word corresponds to a visual grounding module that selects red objects from the scene.

PROGRAMPORT uses a predefined lexicon, manually specified by the programmers. We choose to use a predefined CCG for three main reasons. First, CCG enables a compact definition of a domain-specific semantic parser. For each word in the vocabulary, we only need to annotate 1 or 2 entries. Second, the CCG parser is naturally compositional: given only the syntax and semantic forms for individual words, it is able to parse whole sentences with complex linguistic structures such as nested prepositional phrases. Third, and most importantly, the CCG parser can handle unknown words at test time. In particular, following work on syntactic bootstrapping (Abend et al., 2017; Alishahi & Stevenson, 2008), for a novel word (e.g., *pack the daxy shape into the box*), the parser first "guesses" its syntactic type by ensuring that the entire sentence parses successfully (in this case, the inferred syntactic type is *N/N*), and predicts its semantic form following a prior distribution of $p(semantics|syntax)$, under the current lexicon. In this case, the predicted semantic form is $\lambda x.filter(daxy, x)$.

## 3.2 PROGRAMPORT ARCHITECTURE

PROGRAMPORT executes the parsed programs by following their hierarchical structures. Overall, both *visual grounding* and *action* modules consume outputs from their predecessors and individual concepts parsed by the CCG, instead of the entire language instruction, and produce outputs. By disentangling manipulation in such a structured fashion, our architecture can learn to flexibly leverage vision-semantic priors across a myriad of specific manipulation contexts.

**Visual grounding modules**. The key functionality of a visual grounding module is to localize visual regions corresponding to objects with a given language-specified property (i.e., *filter*(*flower*, ...)). To imbue our manipulation model with semantic understanding of diverse object properties and spatial relationships, including those unseen in the manipulation demonstration data, we borrow from the recent zero-shot semantic segmentation model MASKCLIP (Zhou et al., 2022). Given the input image $\mathbf{x}_t$ and a singular concept $\mathbf{c}$ (e.g., *flower*), we use the image encoder $\mathcal{G}$ and text encoder $\mathcal{H}$ from a pretrained CLIP model to obtain deep features $\mathcal{G}(\mathbf{x}_t) \in \mathbb{R}^{H' \times W' \times D_1}$ and $\mathcal{H}(c) \in \mathbb{R}^{D_2}$, respectively*. Here, we denote the spatial dimensions of the visual features as $H' \times W'$, and the feature dimensions of the visual and language streams as $D_1, D_2$, respectively.

Recall that vanilla CLIP uses attention pooling to pool the visual features at different spatial locations into a single feature vector, and apply a linear layer to align the vector with the language encoding. We adopt a simple modification from MASKCLIP to achieve a similar semantic alignment of features, while preserving the spatiality of $\mathcal{G}(\mathbf{x}_t)$. In particular, we initialize two $1 \times 1$ convolution layers $\mathcal{C}_v$ and $\mathcal{C}_l$ with weights from the penultimate and last linear layers of the CLIP attention pooling mechanism, respectively. The $\mathcal{C}_v$ layer is initialized from the value encoder in the QKV-attention layer (Vaswani et al., 2017), while the $\mathcal{C}_l$ layer is initialized from the last linear layer. Applying them sequentially will project the visual feature into the same space as the language feature. Finally, our grounding module $\mathcal{F}_{grounding}$ generates a dense mask over image regions of concept $\mathbf{c}$ via:

$$\mathcal{F}_{grounding}(\mathbf{x}_t, \mathbf{c}) = \sum \left[ \text{TILE}\left(\mathcal{H}(\mathbf{c})\right) \odot \mathcal{C}_l\left(\mathcal{C}_v\left(\mathcal{G}(\mathbf{x}_t)\right)\right) \right] \in \mathbb{R}^{H' \times W'}, \quad (2)$$

where TILE defines a spatial broadcast operator, in this case for the language embedding vector to the spatial dimensions of the visual features, the $\odot$ is the Hadamard product, and the summation is over the feature dimension $D_2$. This formulation of our semantic module allows us to leverage CLIP's large-scale vision-language pretraining to generalize semantics to open vocabulary categories, including objects and their visual properties. Moreover, since our semantic program has decomposed the input language phrase $\mathbf{l}_t$ into individual visual concepts, we can compose multiple *filter* programs with the intersection operation. Specifically, this is implemented as in NS-CL (Mao et al., 2019) by taking the element-wise min of the generated masks, parsing out image regions that do not satisfy the combination of language descriptors.

**Action modules**. At their core, our action modules extend the architecture from CLIPORT, and finally generate the control parameters $(\mathcal{T}_{pick}, \mathcal{T}_{place})$. In the pick-and-place setting, the visual grounding of the object to be picked (i.e., the mask output by the corresponding *filter* operations) can be directly interpreted as a distribution for $\mathcal{T}_{pick}$. Therefore, in this section, we will focus on how PROGRAMPORT predicts $\mathcal{T}_{place}$ conditioned on $\mathcal{T}_{pick}$, the visual grounding of the target location (e.g., *the orange box*), and the desired relation (e.g., *in*). The corresponding module is the "goal" operator. There are two design principles. First, besides the image and text, the action modules should also be conditioned on the grounding results from visual grounding modules (i.e., a spatial grounding map indicating the position of target objects). Second, we want to prevent overfitting the action module to visual categories presented at training time. Therefore, we extend the architecture for $\mathcal{Q}_{place}$ to take the spatial grounding maps from previous visual modules with the following modifications.

First, we bilinearly upsample or downsample the input masks to the spatial dimensions of different layers within the CLIPort feature backbone, and multiply the interpolated grounding map with these intermediate layers. In contrast to other fusion strategies such as concatenation, this fusion strategy ensures that the grounding map will not simply be ignored by the action modules. Second, we upsample the grounding map to the spatial dimensions of the output action maps, and perform direct element-wise matrix multiplication with the action map. Since the grounding map can be viewed as a mask of the objects being referred to, this multiplication effectively forces the action modules to attend to regions where pick objects and place locations are described in the manipulation instruction. Overall, the end-effector poses in $SE(2)$ can be output by appropriately adapting Eq. (1):

$$\mathcal{T}_{place} = \arg\max_{\tau_i \in \{\tau\}} \text{Up}\left(\mathcal{F}_{grounding}\right) \odot \mathcal{Q}_{place}\left(\tau_i | \mathbf{x}_t, \mathbf{l}_t, \mathcal{T}_{pick}, \mathcal{F}_{grounding}\right), \quad (3)$$

where $\text{Up}(\cdot)$ is an upsampling operator, and $\odot$ is the Hadamard product. More details are available in Appendix A.6. In the simple $SE(2)$ pick-and-place setting under the Transporter or CLIPort framework, simply specifying the locations of the object and its target pose is sufficient, and thus *do(pack, ..., ...)* operates as an identity function.

---

*In practice we use a language prompt instead of a single word, following MASKCLIP (Zhou et al., 2022).

**Example.** Recall that the program for the instruction "pack the blue hexagon in the orange box" is given by *do(goal(filter(filter(hexagon), blue), filter(filter(box), orange), in), pack)*. As shown in Fig. 3, our first set of *filter* modules take the input image and output masks corresponding to "hexagon" shapes and "box" objects. The next stage similarly filters for "blue" and "orange" objects. To compose these adjectives with their corresponding nouns from the previous stage, we take their elementwise min, returning masks corresponding to "blue hexagon" and "orange box", respectively. Next, our *goal* module takes the masks for the referred objects, which correspond to the picked object and its general place region, and outputs a distribution over $SE(2)$ pose for the object that is semantically consistent with the "in" target location. Finally, our *do* module outputs the final control parameters to the robot end-effector. In this case, it contains $\mathcal{T}_{pick}$ from the execution of *filter(filter(hexagon), blue)*, and the $\mathcal{T}_{place}$ output by the *goal* module.

**Training**. Our model can be trained in an end-to-end fashion, given the observation-action tuples $(\mathbf{x}_t, \mathbf{l}_t, \mathbf{a}_t)$, via an imitation learning objective. Note that each action $\mathbf{a}_t$ is itself a tuple described by $(\mathcal{T}_{pick}, \mathcal{T}_{place})$. Thus, given a random observation-action tuple $(\mathbf{x}_t, \mathbf{l}_t, \mathbf{a}_t)$ sampled from some expert demonstration $d_i \in \mathcal{D}$ at time $t$, we fit our model by minimizing the cross-entropy loss:

$$\mathcal{L} = -\mathbb{E}_{\mathcal{T}_{pick}} \left[ \log \operatorname{softmax}(\mathcal{Q}_{pick}((u,v)|\mathbf{x}_t, \mathbf{l}_t)) \right] - \mathbb{E}_{\mathcal{T}_{place}} \left[ \log \operatorname{softmax}(\mathcal{Q}_{place}(\tau_i|\mathbf{x}_t, \mathbf{l}_t, \mathcal{T}_{pick})) \right] \quad (4)$$

## 4 EXPERIMENTS

In this section, we will show that the modularity of PROGRAMPORT enables better vision-language grounding and generalization, which empirically translates to strong zero-shot generalization (i.e., understanding instructions with unseen visual concepts), and compositional generalzation (i.e., understanding novel combination of previously seen visual and action concepts). We also provide validation of our approach on real world experiments in Appendix A.4.

**Dataset**. Our dataset extends the benchmark proposed by CLIPORT (Shridhar et al., 2022a), in turn based on the Ravens benchmark from Zeng et al. (2020). The CLIPORT benchmark consists of 10 language-conditioned manipulation tasks in a PyBullet simulation environment. Tasks are performed by a Universal Robot UR5e with a suction gripper end-effector. Every episode in a task is constructed by sampling a set of objects and properties (such as object type, color, size etc.), and manipulating a subset of the objects to some desired conformation, which is described by a language goal.

**Model details, training, and evaluation**. Our visual grounding modules use the weights from the frozen, pretrained ViT-B/16 CLIP encoder. We train for 200k iterations with AdamW and a learning rate of 0.0002. We use either $n = 100$ or $n = 1000$ training demonstrations, and evaluate on 100 test demonstrations. Our evaluation metric follows Shridhar et al. (2022a) based on the same CLIP encoders, where a dense score between 0 (failure) and 100 (success) represents the fraction of the episode completed correctly. For baselines, we compare to CLIPORT across two settings: the `single` setting, where the model trains and tests on a single task at a time, and the `multi` setting, where the model trains on all tasks at once, and is independently evaluated on each individual task.

### 4.1 ZERO-SHOT GENERALIZATION

To evaluate zero-shot generalization, all of our main results in Table 1 use the "unseen" variants of the tasks, as described in Shridhar et al. (2022a). Concretely, the train and test splits of each task involve different objects and object properties, with minimal overlap. For example, out of 11 total possible colors for objects, only 3 are shared for both train and test episodes. Or for tasks involving real-world objects from the Google Scanned Objects dataset (Downs et al., 2022), a total of 56 household items are split into mutually exclusive sets of 37 train items and 19 test items. Crucially, for a fair comparison, these test language descriptors are also entirely missing from the predefined CCG lexicon, and so semantic goal parsing must rely on the probabilistic approach described in Section 3.1. Our results show that our model effectively grounds object concepts described in language with the visual scene, and with the exception of the align-rope task, outperforms our baselines across the benchmark (Table 1). Align-rope is challenging for PROGRAMPORT because it is hard for the CLIP model to ground abstract spatial concepts used in this task (e.g. *front* and *back*).

**Data efficiency**. Not only do we outperform the baselines by a large margin, but on many tasks, our method does significantly better with 10× fewer demonstrations. For example, on the "packing-shapes" task, our best-performing model trained on 100 demonstrations outperforms its corresponding baseline trained on 1000 demonstrations by 25.0 points.

| # demonstrations | packing-box-pairs | | packing-google-objects-seq | | packing-google-objects-group | | separating-small-piles | | separating-large-piles | |
|---|---|---|---|---|---|---|---|---|---|---|
| | 100 | 1000 | 100 | 1000 | 100 | 1000 | 100 | 1000 | 100 | 1000 |
| CLIPORT (single) | 67.1 | 72.0 | 64.1 | 70.8 | 79.3 | 82.1 | 75.9 | 74.3 | 62.4 | 66.9 |
| CLIPORT (multi) | 77.1 | 72.3 | 66.9 | 71.6 | 77.0 | 81.7 | 65.9 | 63.7 | 54.2 | 59.1 |
| PROGRAMPORT (single) | 72.6 | 78.0 | **78.4** | 79.7 | 79.0 | 81.2 | 76.8 | 77.9 | **63.8** | **69.5** |
| PROGRAMPORT (multi) | **82.3** | **85.7** | 73.5 | **84.0** | **80.7** | **83.9** | **77.0** | **78.8** | 63.0 | 68.2 |

| # demonstrations | align-rope | | packing-shapes | | assembling-kits | | put-blocks-in-bowls | | towers-of-hanoi | |
|---|---|---|---|---|---|---|---|---|---|---|
| | 100 | 1000 | 100 | 1000 | 100 | 1000 | 100 | 1000 | 100 | 1000 |
| CLIPORT (single) | **84.8** | **94.1** | 42.0 | 38.0 | 32.2 | 34.7 | 36.8 | 29.9 | 94.6 | **99.0** |
| CLIPORT (multi) | 81.2 | 70.3 | 38.0 | 32.0 | 30.2 | 26.9 | **55.9** | 46.7 | 75.2 | 69.1 |
| PROGRAMPORT (single) | 77.5 | 86.9 | 52.0 | 54.0 | **40.6** | 42.4 | 38.2 | 38.7 | **97.6** | 98.4 |
| PROGRAMPORT (multi) | 78.4 | 89.1 | **57.0** | **63.0** | 39.2 | **46.8** | 54.2 | **57.7** | 97.0 | 96.3 |

Table 1: **Zero-Shot Generalization.** Training and validation splits for all tasks involve different objects, object properties, or other spatial descriptors.

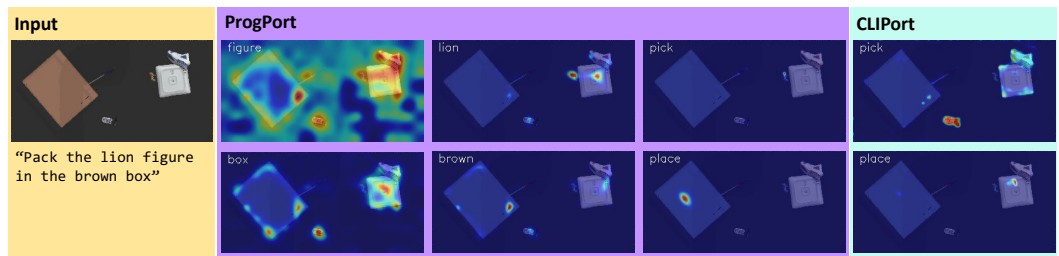

Figure 4: Qualitative pick and place affordances for PROGRAMPORT and CLIPORT. PROGRAMPORT compositionally reasons over independent visual semantics in a zero-shot fashion, correctly picking and placing the target object. CLIPORT identifies both pick object and place location incorrectly.

This data efficiency is derived from our improved grounding properties: by semantically parsing and visually grounding one object concept at a time, we more effectively leverage the pretrained VL model. Moreover, note that the best-performing CLIPORT models for different tasks are mostly the `single` variants. This suggests the zero-shot performance of CLIPORT relies heavily on overfitting to the actions of the specific task, as opposed to leveraging the general visual grounding enabled by CLIP. With PROGRAMPORT, the gap between our `single` and `multi` model variants is much smaller, and indeed the `multi` variants are often the stronger manipulators. PROGRAMPORT thus leverages diverse visual and action information across tasks more effectively, and possesses scalability with respect to both the number of demonstrations as well as the number of tasks.

**Improved vision-language grounding**. Qualitatively, we show more interpretable and robust affordance maps compared to the baseline (Fig. 4). By grounding a single object or property at a time, and then leveraging our DSL operations to compose together these masks, we ground the different concepts. For $\mathcal{T}_{pick}$, the lion figure is extremely small within the visual scene, and positioned in a crowded area; our visual grounding modules are still able to compose the concepts of *figure* and *lion* together in a zero-shot manner, informing an accurate pick affordance. Similarly, for $\mathcal{T}_{place}$, the baseline is confused by the visual similarity between the *box* and the *porcelain plate*. By contrast, our visual grounding modules compose *box* and *brown* together to focus on the left box-like object.

## 4.2 COMPOSITIONAL GENERALIZATION OF VISUAL GROUNDING AND BEHAVIORS

To evaluate compositional generalization, we introduce 4 new tasks which re-combine basic unary object concepts from the original 10 tasks (Table 2). We do not train directly on these tasks; by evaluating on them only, we assess the ability to reason compositionally about the distinct visual properties, spatial relationships, and manipulation primitives learned at training time. For example, the *packing-location-box* task contains instructions of the form "pack the {shape} into the {loc} box", where both `shape` and `loc` are objects and spatial descriptors seen independently throughout various tasks at training time, but now appear in a test-time context that requires the model to reason semantically over their combination. For a full specification of these tasks, please refer Appendix A.2.

| | packing-color-box | | packing-location-box | | separating-location-piles | | pushing-shapes | |
|---|---|---|---|---|---|---|---|---|
| | 100 | 1000 | 100 | 1000 | 100 | 1000 | 100 | 1000 |
| CLIPORT (single) | 23.0 | 24.0 | 11.0 | 13.7 | 48.1 | 43.8 | 2.4 | 3.2 |
| CLIPORT (multi) | 19.0 | 18.2 | 10.0 | 8.3 | 39.9 | 41.0 | 2.7 | 7.6 |
| PROGRAMPORT (single) | 64.0 | 68.0 | 56.0 | 62.5 | **69.2** | **72.4** | 32.3 | 36.9 |
| PROGRAMPORT (multi) | **72.0** | **76.3** | **62.0** | **67.1** | 68.4 | 70.5 | **36.6** | **40.2** |

Table 2: **Compositional Generalization.** Models can learn different object shapes, colors, and manipulation properties from different tasks. We show that only PROGRAMPORT hierarchically composes these learned concepts and behaviors together to perform well in more complex tasks.

| Method | Semantic Supervision | packing-shapes | | packing-google-objects-seq | | assembling-kits | | separating-small-piles | | put-blocks-in-bowls | |
|---|---|---|---|---|---|---|---|---|---|---|---|
| | | 100 | 1000 | 100 | 1000 | 100 | 1000 | 100 | 1000 | 100 | 1000 |
| PROGRAMPORT (single) | MaskCLIP | 52.0 | 54.0 | 78.4 | 79.7 | 40.6 | 42.4 | 76.8 | 77.9 | 38.2 | 38.7 |
| | GT | **97.0** | **98.0** | **95.8** | **95.9** | **92.0** | **93.4** | **80.6** | **86.4** | **97.2** | **96.3** |
| PROGRAMPORT (multi) | MaskCLIP | 57.0 | 63.0 | 73.5 | 84.0 | 40.6 | 42.4 | 77.0 | 78.8 | 54.2 | 57.7 |
| | GT | **96.0** | **98.0** | **96.2** | **96.4** | **91.0** | **94.5** | **81.2** | **87.2** | **99.3** | **97.7** |

Table 3: **Disentanglement of visual semantics and action.** Replacing MASKCLIP grounding maps with ground truth segmentation maps of objects and targets yields near-perfect manipulation.

**Length-based generalization to new action settings**. Our fourth generalization task (pushing-shapes) is much more difficult, involving language instructions of the form "push the {color} {shape} into the {location} {color} square". Note that this task evaluates both compositional generalization and "length generalization". The latter assesses the ability of the model to generalize beyond the assumption of at most one visual descriptor per object in similar training tasks. Furthermore, this task requires the model to zero-shot generalize to entirely new actions; only pushing of simple, regularly-structured blocks is learned at training time, whereas shapes are much larger, irregular, and have different friction properties. PROGRAMPORT exhibits much better performance here (+32.6 mean reward), compared to the CLIPORT baseline, which fails entirely.

### 4.3 ABLATION: DISENTANGLEMENT OF ACTION AND PERCEPTION

To illustrate the disentanglement of action and perception in our modular design, we perform an additional ablation study. Specifically, we train our model normally on all 10 main tasks from Section 4.1, and at test time, we choose 5 diverse tasks out of the 10. For each test episode, we replace the attention maps generated by the semantic modules with ground truth segmentation maps corresponding to the picked objects and place locations. This effectively poses the question: given a perfect perception of the scene, how would our action modules behave? A properly disentangled model would be able to leverage the noiseless ground truth visual grounding and decide the appropriate actions to perform. As apparent in Table 3, performance across all tasks improves dramatically when provided the ground truth information. This suggests that our action modules do not overfit to task-specific shortcuts when optimizing the imitation objective, but actually learn to leverage the attention maps provided by the semantic modules in a disentangled manner. Such disentanglement has also been translated to improved compositional generalization for our model.

## 5 RELATED WORK

**Learning language-guided robot behavior**. Many prior works have explored enhancing robotic behavior through language-conditioned imitation learning (Lynch & Sermanet, 2021; Stengel-Eskin et al., 2022; Stepputtis et al., 2020) and reinforcement learning (Andreas et al., 2018; Luketina et al., 2019; Misra et al., 2017). With the rise of powerful transformer-based architectures, recent works have used large-scale pretrained vision-language representations to improve performance for tabletop manipulation (Shridhar et al., 2022a), visual navigation (Khandelwal et al., 2022), and real-world skill learning and planning (Ahn et al., 2022; Huang et al., 2022). For example, CLIPORT shows that by integrating embeddings from the CLIP VL model into the Transporter manipulation framework (Zeng et al., 2020), a robot can better generalize to manipulating new objects in a zero-shot manner.

However, in contrast with our method PROGRAMPORT, these works do not leverage the semantic structure in natural language to ground individual visual and action concepts. Rather, prior works use the pretrained language model as a monolith that consumes the entire language input and outputs a latent embedding. This embedding is not grounded to individual concepts within the language inputs, which leads to poor generalization when faced with novel (combinations) object attributes.

**Neurosymbolic learning**. Prior works have shown neurosymbolic learning, which refers to the integration of deep neural networks for pattern recognition and symbolic structures such as programs for reasoning, as an effective mechanism for enforcing VL grounding (Mao et al., 2019; Wu et al., 2019). Mao et al. (2019) is a representative work that learns to jointly disentangle independent visual concepts and parse natural language sentences, from natural supervision in VQA. The learned concepts can be composed for novel downstream tasks via programmatic (re)composition of neural network modules, enabling zero-shot and compositional generalization. More generally, a structured approach involving neural perception of vision and language, and higher-order symbolic reasoning has appeared previously (Li et al., 2020; Nottingham et al., 2021; Sun & Alexandre, 2013). Beyond improved VL-grounding, this approach has also translated to improved performance and data efficiency on many downstream tasks, such as reinforcement learning (Cheng et al., 2019; Verma et al., 2019), scientific discovery (Cranmer et al., 2020; Shah et al., 2020) and robotics (Zellers et al., 2021).

Our work is similarly inspired to combine structured representations with deep learning, in the new setting of robot manipulation. However, instead of learning a small set of visual concepts, we leverage pretrained VL models to cover a larger, open-ended set of concepts, and improve our zero-shot capabilities. Moreover, we ground not just visual concepts with their language descriptors, but also abstract actions for robotic primitives, while preserving compositional generalization capability.

**Vision-language (VL) grounding**. The goal of VL grounding is to identify correspondences between visual input and descriptors in natural language. Common tasks include visual question answering (Antol et al., 2015), VL navigation (Anderson et al., 2018), or VL manipulation (Shridhar et al., 2022a;b). Recent works show that the CLIP (Radford et al., 2021) model trained on 400M image-caption pairs from the web possesses zero-shot VL grounding capabilities for a wide array of visual concepts (Kim et al., 2021; Shen et al., 2022). In robotics, this has translated to grounding of robotic affordances with natural language in models such as CLIPORT (Shridhar et al., 2022a).

However, we find that the naïve approach taken by these works, which leverage only the frozen, pretrained CLIP embeddings, cannot successfully disentangle task-agnostic visual grounding with task-specific action policies. Other works show that this naïve integration of pretrained VL models also affects grounding abilities in 3D space (Thomason et al., 2022), or temporal settings such as video understanding (Le & Hoi, 2020). In particular, VL grounding is often adversely affected by end-to-end training (e.g. via an imitation objective), and the resultant models are unable to reason about novel concepts independently, or compose them de novo for more complex tasks (Lin et al., 2022). This shortcoming is addressed by PROGRAMPORT through a program-based modular approach.

# 6  CONCLUSION

In this work, we propose PROGRAMPORT, a program-based modular framework for robotic manipulation. We leverage the semantic structure of natural language to programmatically ground individual language concepts. Our programs are executed by a series of neural modules, dedicated to learning either general visual concepts or task-specific manipulation policies. We show that this structured learning approach disentangles general visual grounding from policy learning, leading to improved zero-shot and compositional generalization of learned behaviors.

Future work of PROGRAMPORT includes generalization to more free-form natural language instructions and challenging scenarios where the syntactic context is not sufficient to predict semantics of novel words. Alternate techniques such as extending human-annotated CCG banks (Bisk & Hockenmaier, 2013; Zettlemoyer & Collins, 2012), and leveraging the wealth of pretrained language-to-code models (Chen et al., 2021) can be a concrete path to alleviating this issue. Better representations that can capture interactions between robots and objects (e.g. grasping poses for different objects) is another important extension. Finally, subsequent work may also consider jointly learning and leveraging pretrained representations that are more suitable for grounding interaction modes between robots and objects, especially for manipulating difficult articulated and deformable objects in our 3D world (Shridhar et al., 2022b).

## REPRODUCIBILITY STATEMENT

Our simulated dataset details are provided in the Appendix A.2. The most salient training details and hyperparameters are presented in Section 4 of the main text. DSL details can be found in Appendix A.1 and architecture details are available in Appendix A.6. Our code will also be publicly available.

## ACKNOWLEDGMENTS

This work is in part supported by the Stanford Institute for Human-Centered AI (HAI), Analog Devices, JPMC, and Salesforce. This work is also supported by the Ministry of Science and Technology of the People´s Republic of China, the 2030 Innovation Megaprojects "Program on New Generation Artificial Intelligence" (Grant No. 2021AAA0150000, No. 2022ZD0161700), as well as a grant from the Guoqiang Institute, Tsinghua University.

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

# A APPENDIX

## A.1 DOMAIN-SPECIFIC LANGUAGE

We detail here the operations (Table 4) and types (Table 5) contained within our DSL. The implementation of *filter*, *do* and *goal* are given within the main text. The remaining operations are parameter-free. In particular, *Scene* returns a uniform attention map over the visual input; *ObjUnion* and *ActionConcat* take the elementwise max of their input grounding or action maps, respectively.

| Operation | Signature | Semantics |
|---|---|---|
| Scene | () ⟶ Object | Return all objects in visual scene. |
| Filter | (Object, ObjProp) ⟶ Object | Return all objects satisfying a vision-language property. |
| Relate | (Object, Object, ObjRel) ⟶ Object | Return all objects satistfying a relation between target and reference objects. |
| Do | ({Goal}, Action) ⟶ Plan | Perform an action according to a set of goals. |
| Goal | (Object, Object, ObjRel) ⟶ Goal | Identify a manipulation goal for an object and target relation. |
| ObjUnion | (Object, Object) ⟶ Object | Return the union of two (sets) of objects. |
| ActionConcat | (Plan, Plan) ⟶ Plan | Jointly perform two sets of actions. |

Table 4: All operations in our DSL for vision-language manipulation.

| Type | Example | Semantics |
|---|---|---|
| Object | Block, Rope, etc. | Manipulation objects. |
| ObjectProp | Red, Dark, etc. | Object properties. |
| ObjRel | In, On, etc. | Object relationships. |
| Goal | - | Manipulation target such as distributions over SE(2) poses. |
| Plan | - | End-effector control parameters in SE(2). |
| Action | Pack, Push, etc. | Manipulation primitives. |

Table 5: All types in our DSL for vision-language manipulation.

## A.2 DATASET DETAILS

Our dataset is based off the Ravens benchmark proposed in Zeng et al. (2020), and its adaptation in Shridhar et al. (2022a). In particular, for our manipulatives, we use the same set of 20 synthetic shapes and 56 Google Scanned Objects. For modifying visual appearance, we use the same set of 11 colors . Most importantly, we use the same split of "seen" and "unseen" objects and colors, training on the "seen" splits and testing on the "unseen" splits. These splits are mutually exclusive for the manipulatives; only 'red', 'green' and 'blue' are shared between splits for the colors.

In the remainder of this section, we detail the specific tasks used within our benchmark, including visual scene setup and the success metric used to compute the results in Table 1, Table 2 and Table 3.

### A.2.1 BASIC TASKS

**Packing Box Pairs.** Given small boxes of varying colors, and a larger, open-topped brown box, the goal is to *tightly* pack all small boxes of two specified colors into the large brown box. Sizes of all boxes are randomized, and there will be an exact number of the specified small boxes to tightly pack the larger box, in addition to other distractor small boxes. Occasionally, there will be boxes of only

one of the two specified colors; the agent must learn to ignore the missing color and pack the boxes with the one present color.

Success is the fraction of total volume of specified color blocks in the scene that end up packed inside the brown box.

**Packing Google Objects Seq.**    *At each timestep*, given a number of Google scanned objects and an open-topped brown box, the goal is to place a specified object into the box. Each episode is composed of at least 1 and at most 5 timesteps.

Success is determined by the total volume of all specified objects within the brown box up to the given timestep, divided by the volume of all specified objects.

**Packing Google Objects Group.**    Given a number of Google scanned objects from at least two different categories, as well as an open-topped brown box, place all objects of a specified category into the brown box.

Success is determined by the fraction of volume of specified objects placed within the box,

**Separating Small Piles.**    Given 50 small blocks and two square zones of different colors, positions and poses, push all the blocks into the zone with a specified color.

Success is determined by the fraction of total blocks within the correct zone.

**Separating Large Piles.**    Given 15 large blocks and two square zones of different colors, positions and poses, push all the blocks into the zone with a specified color.

Beyond size changes, the large blocks also have more mass and lateral friction compared to their small counterparts.

Success is determined by the fraction of total blocks within the correct zone.

**Align Rope.**    Given a soft-body rope formed from 20 articulated beads and the outline of a 3-sided square, the goal is to connect the two endpoints of the rope to two adjacent corners of the square. The language instruction can specify one of four possible configurations:

1. From front left tip to front right tip
2. From front right tip to back right corner
3. From front left tip to back left corner
4. From back right corner to back left corner

Success is represented by the degree to which the bead poses after manipulation match the line segment between the two adjacent corners specified in the language goal.

**Packing Shapes.**    Given 4 distractor shapes and 1 specified shape, as well as a open-topped brown box, the goal is to put the specified shape into the brown box. Shapes are all unique.

Success is binary; 1 if the shape is inside the boundaries of the box, and 0 otherwise.

**Assembling Kits.**    Given 5 shapes and a "kit" of 5 similarly-shaped holes of varying poses, the goal is to precisely place each shape into its corresponding hole. Shapes are *not* unique; in the event that more than one of the same shape exists in the field, the language goal will specify a one-to-one mapping between shapes and holes, e.g. "put the orange star into the left star shape hole."

Success is determined by the fraction and degree to which placed shapes match the poses of the shape holes.

**Put Blocks in Bowls.**    Given a number of blocks of varying colors, and a (possibly different) number of bowls of (possible different) varying colors, the goal is to place all blocks of a specified color into some bowl of a specified color. Note that each bowl will fit at most one block, and it is guaranteed that there are a sufficient number of bowls to achieve the goal.

Success is determined by the fraction of blocks of the specified color being within any bowl of the specified color.

**Towers of Hanoi.** *At each timestep*, identify and move a ring shape from one of three possible pegs to another specified peg. Each episode contains 7 such timesteps, which is the ideal solution for 3-ring Towers of Hanoi. The sequence of moves is deterministically composed apriori using a DFS solver.

Success is determined by the fraction of correct ring placements over timesteps.

**Packing Prepositions.** Given 4 distractor shapes and 1 specified shape, as well as *two* brown boxes, the goal is to place the specified shape into the box with the specified spatial relationship. This spatial relationship is given *relative* to one of the 4 distractor shapes. For example, "pack the flower into the brown box *left of* the star."

Success is binary; 1 if the shape is inside the boundaries of the correct box, and 0 otherwise.

### A.2.2 COMPOSITIONAL TASKS

After training on basic tasks, we evaluate the baselines and our model on the four compositional tasks described below. In addition to the visual scene setup and success metric, we also describe their compositional nature and from which basic tasks we expect the model to learn the requisite building block behaviors.

**Packing Color Box.** Given 4 distractor shapes and 1 specified shape, as well as *two* boxes of different colors, the goal is to place the specified shape into the box with the specified color. Shapes are still unique.

Shapes are seen in the "packing-shapes" task and colors are seen in the "packing-box-pairs" task, as well as the 'assembling-kits," "put-blocks-in-bowls," and "separating-small/large-piles" tasks.

Note that at training time, the model only ever packs objects into a single brown box. Hence, this task assesses the model's ability to disentangle and separately ground the packing action and the visual properties of the box.

Success is binary; 1 if the shape is inside the boundaries of the correct box, and 0 otherwise.

**Packing Location Box.** Given 4 distractor shapes and 1 specified shape, as well as *two* brown boxes, the goal is to place the specified shape into the box with the specified spatial relationship. For example, "pack the flower into the *left* brown box." Shapes are still unique.

Shapes are seen in the "packing-shapes" task and spatial descriptors are seen in the 'assembling-kits" task.

Again, the model only ever packs objects into a single brown box at training time. This task assesses the model's ability to compose spatial descriptors with the packing action.

Success is binary; 1 if the shape is inside the boundaries of the correct box, and 0 otherwise.

**Separating Location Piles.** Given 50 small blocks and two square zones of the same color and size, but different positions and poses, push all the blocks into the zone with a specified spatial relationship. For example, "push the pile of purple blocks into the left square."

The small blocks and pushing action are present in the "separating-small-piles" task and spatial descriptors are seen in the 'assembling-kits" task.

Akin to the "packing-location-box" task, this task assesses the model's ability to compose spatial descriptors with the pushing action.

Success is determined by the fraction of total blocks within the correct zone.

**Pushing Shapes.** Given 4 distractor shapes and 1 specified shape, and two square zones of the same size, but different colors, positions and poses, the goal is to push the specified shape into the

specified zone. Zones also have a spatial descriptor; for example, "push the green ring into the left blue square." Shapes are no longer unique, but rather combinations of color and shape are unique (e.g. there may be two "ring" shapes in the scene, but only one "green ring" shape.)

The pushing action is present in the "separating-small/large-piles" tasks, spatial descriptors are seen in the 'assembling-kits' task and shapes are seen within the "packing-shapes" task.

This task is highly challenging. The shapes pushed at test time are distinct from those seen at training time in "packing-shapes," and the pushing action is only learned for blocks in the 'separating-small/large-piles" tasks. Hence, we test for zero-shot and compositional generalization to novel objects and visual properties, but also to novel actions. Finally, introducing both spatial and color descriptors to the target square zones also adds a length-based dimension to the generalization properties.

**Packing Nested Prepositions.** Given 4 distractor shapes and 1 specified shape, as well as *two* brown boxes, the goal is to place the specified shape into the box with the specified spatial relationship. This spatial relationship is given *sequentially*, *relative* to 2 of the 4 distractor shapes. For example, "pack the flower into the brown box *left of* the star *right of* the diamond." Shapes are still unique.

Shapes are seen in the "packing-shapes" task and relative spatial descriptors are seen in the 'packing-prepositions" task.

Again, the model only ever packs objects into a single brown box at training time. This task assesses the model's ability to compose relative spatial descriptors alongwith the packing action.

Success is binary; 1 if the shape is inside the boundaries of the correct box, and 0 otherwise.

## A.3 Experiments with Prepositional Phrases

Task details are available in Appendix A.2.

|  | packing-prepositions | packing-nested-prepositions |
|---|---|---|
| CLIPort | 28.0 | 17.0 |
| ProgramPort | 45.0 | 31.0 |

Table 6: **Experiments with *relate* operator.** *packing-prepositions* evaluates zero-shot generalization whereas *packing-nested-prepositions* evaluates compositional generalization.

## A.4 Real World Experiments

We collect 40 demonstrations of tabletop pick-and-place tasks using a Franka Panda robot arm with a parallel gripper end-effector. Pick objects are common household objects including tape measure, tennis balls, spoons, as well as plastic fruits such as bananas, pears, peaches, etc. These are placed into plates, bowls and blocks of different colors (red, blue, green, purple etc.). For example, a language goal may take on the form "pack the banana in the red plate." Success criteria is the same as the "packing-shapes" task in simulation. We train on 30 demonstrations, and hold out the other 10 demonstrations for validation. For test evaluation, we use 10 demonstrations for in-distribution pick objects and place locations, and another 10 with out-of-distribution pick objects and place locations. Test results are shown in Table 7 and visualized in Fig. 5.

|  | packing-objects-seen | packing-objects-unseen |
|---|---|---|
| ProgramPort | 70.0 | 80.0 |

Table 7: **Real world packing objects task.** The *seen* suffix involves new instructions containing the same objects seen in the training set, whereas the *unseen* suffix evaluates zero-shot generalization to new instructions containing novel objects.

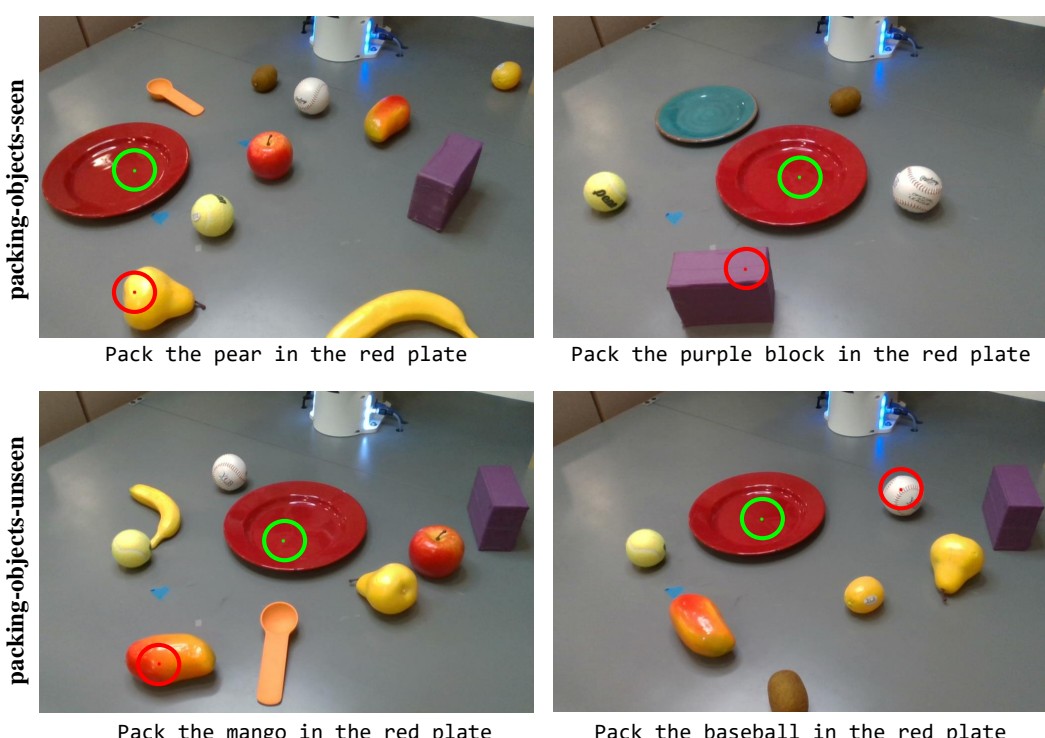

Figure 5: **Real world packing-shapes task**. Pick locations on objects are highlighted in green, and place locations are highlighted in red.

## A.5    ADDITIONAL AFFORDANCE VISUALIZATIONS

We present qualitative examples of PROGRAMPORT's and CLIPORT's attention mask across four different tasks in Figures 6, 7, 8, 9. PROGRAMPORT outputs masks over every independent visual concept, while CLIPORT only outputs masks over the final pick and place locations. For all tasks shown, PROGRAMPORT is able to more accurately localize the target locations compared to CLIPORT.

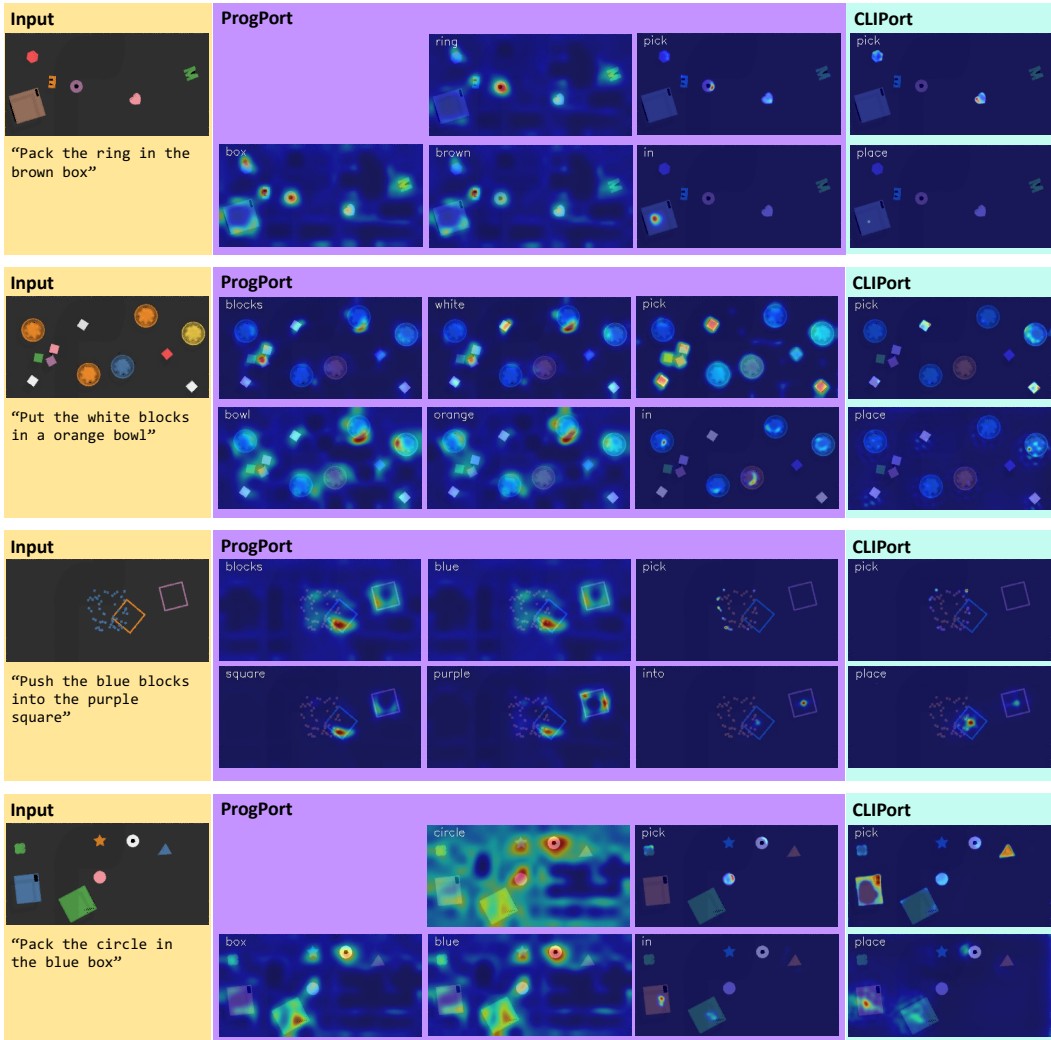

Figure 9: **Qualitative affordance evaluations for different zero-shot and compositional tasks**.

## A.6 MODEL DETAILS

In this section, we also illustrate and further describe the architectures of our visual grounding and action modules.

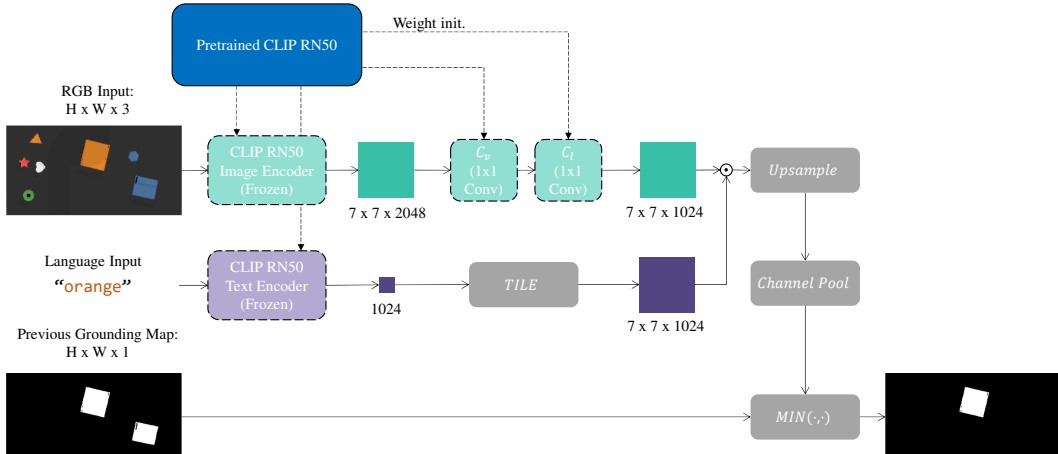

Figure 10: **Visual grounding module example: *filter*.**

Our visual grounding modules leverage pretrained CLIP embeddings via a layer-weight initialization scheme. Formally, CLIP is trained via an embedding-level contrastive objective on 400M image-caption pairs, which we represent as tuples $(\mathbf{x}_t, \mathbf{l}_t)$. CLIP is comprised of an image encoder (ResNet-50) $\mathcal{G}$ and a text encoder (transformer) $\mathcal{H}$, which featurize their respective components in the input, yielding a visual feature map $\mathcal{V}$ and a text embedding $\mathbf{u}$. To generate a visual embedding vector $\mathbf{v}$, a spatial weighted-sum of $\mathcal{V}$ is performed via qkv-style attention (Vaswani et al., 2017), followed by a linear layer $\mathcal{F}$:

$$\mathbf{v} = \mathcal{F}\left(\sum_{i,j} \mathrm{softmax}\left(\frac{\mathrm{Emb}_q\left(\mathrm{AvgPool}\left(\mathcal{V}\right)\right) \cdot \mathrm{Emb}_k(\mathcal{V}_{i,j})^T}{C}\right) \mathrm{Emb}_v(\mathcal{V}_{i,j})\right) \quad (5)$$

where $\mathrm{Emb}_{q,k,v}(\cdot)$ are linear embedding layers and $C$ is some fixed scalar. Image and text embeddings $\mathbf{u}, \mathbf{v}$ are then orchestrated to the same representation latent space via CLIP's contrastive training objective.

To obtain an attention map over $\mathbf{x}_t$ representing the language concepts described in $\mathbf{l}_t$, we need to maintain the spatiality of the visual feature map $\mathcal{V}$. To this end, we remove the summation over pixels $i, j$, and introduce two $1\times1$ convolution layers, $\mathcal{C}_v$ (initialized with the weights from linear embedding layer $Emb_v$) and $\mathcal{C}_l$ (initialized with the weights from the linear layer $\mathcal{F}$). This is illustrated in Fig. 10.

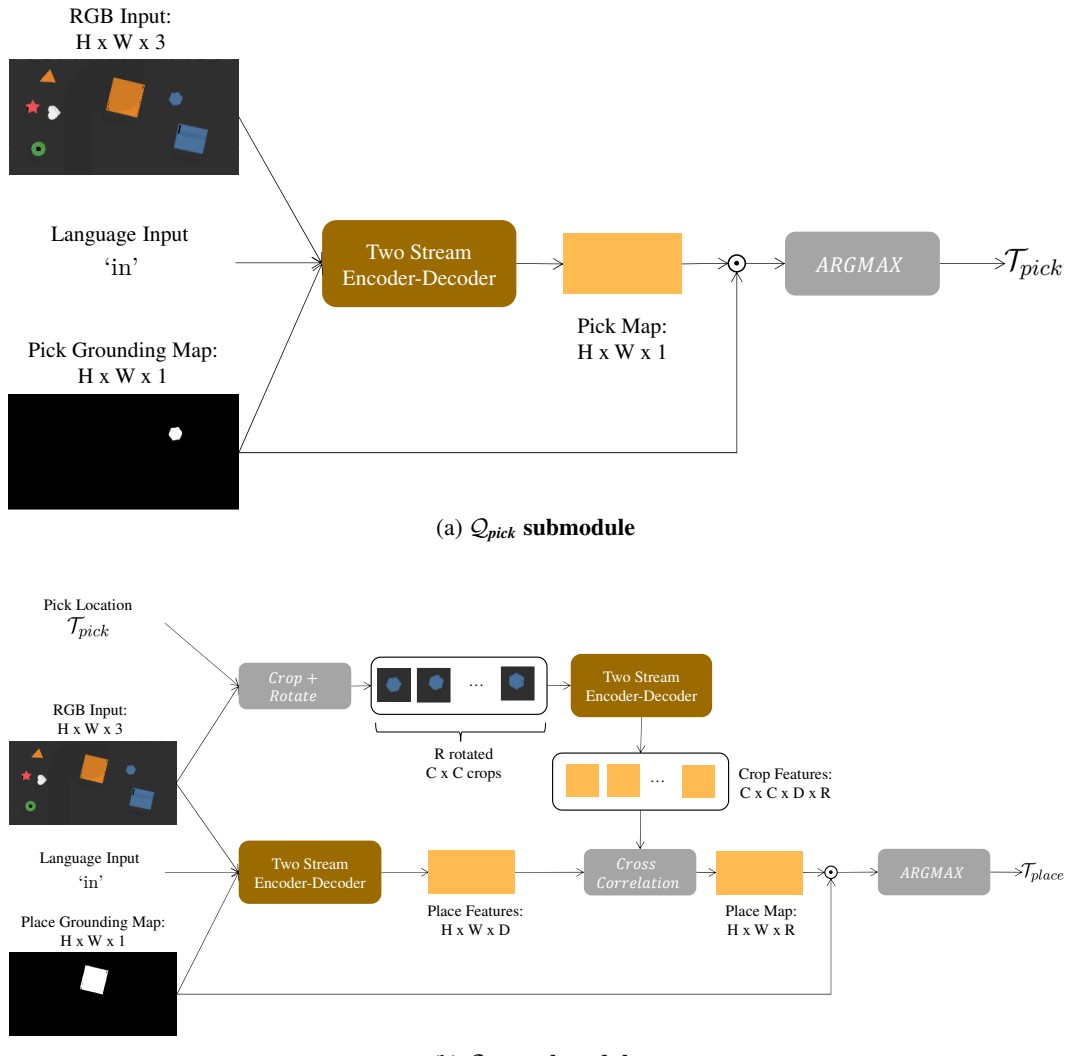

(a) $\mathcal{Q}_{pick}$ **submodule**

(b) $\mathcal{Q}_{place}$ **submodule**

Figure 11: **Action module example: *goal.***

The two-stream FCN encoder-decoder architecture used in our action modules borrow heavily from CLIPort. However, we find that simply treating the pick or place grounding maps as an additional fourth channel to the RGB visual input results in the decoder layers ignoring the visual semantics represented by these grounding maps. To solve this issue, we first bilinearly upsample or downsample the grounding map to the spatial resolution of different layers of the decoder. We then directly multiply these grounding maps with the outputs of the decoder layers, explicitly encoding the notion of attention into these latent features.

