# OpenReview forum: "Programmatically Grounded, Compositionally Generalizable Robotic Manipulation"
_ICLR.cc/2023/Conference — ICLR 2023 notable top 25%_

### Official Review · Reviewer_cgai · 2022-10-23

**Confidence:** 4
**Correctness:** 4
**Technical Novelty And Significance:** 3
**Empirical Novelty And Significance:** 3
**Recommendation:** 8

**Clarity, Quality, Novelty And Reproducibility:**

Clarity: The paper is well-written
Quality: The technique is sound and the experimental results are good.
Novelty: The overall pipeline design is novel.

**Strength And Weaknesses:**

Strength:

1. The paper is well-written and easy to follow.

2. The proposed framework is novel, and has great performance.

Weakness:

The idea of this paper is somewhat novel, but not entirely refreshing. It seems a combination of different known techniques: CLIP/MaskCLIP, minor modification of CLIPort.
With the modularization and structured CCG, zero-shot generalization ability and compositional generalization ability is quite predictable.
Nevertheless, putting all these techniques together and proves the performance gain are also contributions for community.

**Summary Of The Paper:**

The paper proposes a vision-language-action grounding framework for robot manipulation, with the guidance of a domain specific language Combinatory Categorial Grammar (CCG). CCG first parse the natural language to more structured program, and associate the objects and targets via the grounding module. Then it produces the action parameters via action module. This method has advantages over many tasks than the baseline methods in zero-shot generalization and compositional generalization.

**Summary Of The Review:**

It is a good experimental work, with sound techniques and good performance.
The paper is well-written and well-motivated.
I think it is of quality to publish.

---

### Official Review · Reviewer_LWwk · 2022-10-24

**Confidence:** 3
**Correctness:** 4
**Technical Novelty And Significance:** 3
**Empirical Novelty And Significance:** 3
**Recommendation:** 8

**Clarity, Quality, Novelty And Reproducibility:**

The work is relatively clear though a significant part of the description of the method relies on familiarity with the CLIPort paper and methods therein. Nevertheless, I believe that the information in the CLIort paper makes up for what's missing in this paper.
The web page referenced and heat maps are great visualizations of what was extracted from the VL model and the quality of performance achieved. Hopefully the code will be made available as an accurate reference for how training is done (which I felt is a bit vague and more detail would have helped).

Some specific things that I found unclear are mentioned in the previous section.

**Strength And Weaknesses:**

This paper is an exemplary case of harnessing general-purpose modeling of the world (scenes and their descriptions) to solve an engineering problem within a limited domain (a small set of manipulation tasks involving two objects / entities) by combining the strengths of the general-purpose model and programmable human knowledge of the task breakdown so that the the ML modeling that remains to be performed is focused on the aspects of the solution that are still missing.

One might say that the model architecture in this study benefits from an "unfair advantage" over previous work in that the designers used their understanding of manipulation tasks to better constrain the scope of what the VL model contributes to the architecture. I see this as good engineering work for solving robotic manipulation in settings where the set of entities to be manipulated is open ended.

Perhaps a weakness of this line of work, as far as I understand it, is that there exist robotic tasks in which a vision + language model may have more to share with an action module than a set of masks. For example, a situation where the text cound be used to describe a novel task (or any other case where the DSL proposed is too contraining). Such enhancement would require making a new custom parser and extracting new maps / embeddings from the VL model.

- I found equation 2 non-intuitive (due to lack of familiarity with what the Cl and Cv layers signify).
- In some cases it seems that the pick action is referred to as a 2D (pixel location) vector whereas in others it seems to be a location+rotation.
- I am not clear on where the observed action is used in eq 4 (is it the (u,v) and tau_i?)
- Is it correct to say that the only information passed between the VL model and the action model is two heatmaps and a task type (pack or push)? If so, would the authors like to discuss whether the model is missing out on some additional information that the VL model might have?

- An experiment showing transfer of the skills learned in simulation to a real robot would have been super.


**Summary Of The Paper:**

The paper is an improvement-through-specialization upon prior work in robotic manipulation where Vision-Language (VL) models allow for manipulation of novel objects or objects that are specified using novel characteristics (not used in training).

The study makes use of a previously described simulation-based robotic manipulation framework (and data?) to train a novel model architecture. Whereas a previous ML modeling study in robotic manipulation (called CLIPort) trained an action module directly from image and text embeddings, this study is able to train for better performance and better generalization by taking advantage of the constrained nature of the available robotic manipulation tasks within a limited framework. This is performed by limiting the space of task descriptions to a Domain-Specific Language (DSL) that can be easily parsed to extract physical entities and their descriptors plus the required task type from a limited (trained-for) set of available manipulation tasks.

In this manner - the study takes advantage of the stength of VL models to narrow down (mask) the relevant locations for start and end of manipulation tasks (pick A and place in B where A and B can be rich object and location entity descriptions). Specifically, by parsing the commands into task, pick and place entities with a DSL, I believ that they avoid mixing of characteristics between the two entities (which sometimes occures with VLs) and they can use the features that describe objects as filters that progressively limit the possible locations within the image (e.g. "blue box" is parsed into a filter for blue and a filter for box whose output can be intersected to produce an accurate location mask). The action module is then trained on the processes masks and is thus disentangled from the task of understanding where the task entities are.

The results show very significant improvements across several manipulation tasks in terms of performance, generalization to unseen entities and across task types while also exhibiting better data efficiency (within the constrained domain) compared with the previous work.


**Summary Of The Review:**

The paper presents a novel way of training a robotic manipulation model with help from a Vision Language model for localization of novel objects. It solves a problem encountered in a previous study where it seems that the model learned was unable to efficiently disentangle aspects of scene understanding from aspects of action generation. This was done by limiting the text that describes the task to a domain specific language (as opposed to NLP) and parsing it with a parser that is tailored to the limited task domain.
The architecture is able to efficiently learn to manipulate unseen objects or seen objects in manipulation tasks where they were not observed before. An ablation experiment shows that the action module performs nearly perfectly when the VL outputs are replaced with ground-truth segmentation.

---

### Official Review · Reviewer_x8wi · 2022-10-24

**Confidence:** 3
**Clarity, Quality, Novelty And Reproducibility:** See above
**Correctness:** 3
**Technical Novelty And Significance:** 3
**Empirical Novelty And Significance:** 3
**Recommendation:** 8

**Strength And Weaknesses:**

## Paper strengths and contributions

**Motivation and intuition**
- The motivation for making visual grounding tasks more generalizable by utilizing compositional program-based modules is convincing and intuitive.
- The limitations of prior works are explicitly mentioned, which strengthens the motivation of this work.

**Technical contribution**
The proposed an action module that is conditioned on the grounding map and the extended architecture from CLIPort seem effective for preventing overfitting.

**Clarity**
- The overall writing is clear. The authors utilize figures well to illustrate the ideas. Figure 1 clearly shows the tasks that this paper deals with and provides task examples to explain the figure.
- The authors explain their intuitions and reasons well when introducing new ideas.

**Related work**
- The authors give a clear description of related prior works in learning language-guided robot behaviors, neurosymbolic learning, and vision-language grounding.
- The authors provide clear comparisons that highlight the difference between their work and related previous work, which makes it easier to follow for those who are not familiar with the background.

**Ablation study**
Ablation studies presented in Table 3 justify the effect of disentangling the visual semantic and the action module, which can be translated into compositional generalization for the model. It is helpful to understand the design choice.

**Experimental results**
The experimental results demonstrate the generalization performance of the proposed model.

## Weaknesses and questions

**Clarity**
- It is unclear to me how we can predict p(semantic/syntax) in the last part of Section 3.1. I wonder what the intuition is to determine in what situation the conditional probability p should be higher given an inferred syntactic type (N/N).
- It is a little difficult for me to follow the description of the architecture of the visual grounding module in Section 3.2. Providing a diagram explaining the architecture presented in previous works, the modification made in this work, and I/O features between modules and layers, would be helpful.

**Summary Of The Paper:**

This paper addresses the problem of solving visual-language (VL) grounding tasks by leveraging pre-trained VL models. It is motivated by the fact that conventional methods are not data-efficient in the training stage and have poor generalization to unseen objects and tasks. To this end, this paper proposes a framework that first parses an instruction into a program based on a domain-specific language and then composes functional modules, a visual grounding module, and an action module, to execute the program. The experiments show that the proposed framework outperforms a baseline based on a two-stream fully connected architecture in terms of both generalization and data efficiency. The ablation studies suggest that the action module works with ground segmentation maps and does not overfit to individual tasks in a disentangled manner of action and perception. I am leaning toward accepting this paper since it studies a promising research direction and presents a reasonable framework to address the problem with supportive experimental results.

**Summary Of The Review:**

I am leaning toward accepting this paper since it studies a promising research direction and presents a reasonable framework to address the problem with supportive experimental results.

---

### Official Review · Reviewer_dRES · 2022-10-25

**Confidence:** 5
**Correctness:** 3
**Technical Novelty And Significance:** 2
**Empirical Novelty And Significance:** 2
**Recommendation:** 5

**Clarity, Quality, Novelty And Reproducibility:**

Clarity: The manuscript should rephrase the following statement: "To enable consistent reference to concepts across diverse tasks and scenes, many such models also take as input natural language instructions." Actually, natural language references are generally considered to be inconsistent across diverse tasks, scenes, and locales. However, natural language *does* (typically) provide an abstract goal specification in a way that facilitates better communication with other agents (e.g., humans).

Clarity: The manuscript should make exceedingly clear about which aspects of the approach are learned, frozen, fine-tuned, or are deterministic.

Novelty: The manuscript provides a principled engineering approach for vision-language manipulation, using a combination of programmatic functions via deterministic models and layer-weight initialization strategies via pre-trained models. I am concerned that the manuscript provides neither compelling insights from a new methodological approach nor does it provide any intuition about learned representations.

**Strength And Weaknesses:**

(Strengths)

Provides a framework for combining vision-language models with structured grammar, without further fine-tuning.

(Weaknesses)

Section 1: Many works seek to use pre-trained vision-language models as a frozen semantic prior. It is true that fine-tuning is usually regarded as being too costly, may yield catastrophic forgetting, or simply dismisses the rich experience obtained through the vision-language model's pretext task. However, layer-weight initialization from the pre-trained vision-language model is not without its problems either — chief of which is that the vision-language model is likely not aware of the downstream task. Discussion is missing in the manuscript about why this is not a problem for the proposed approach.

Section 2: This formulation seems to assume that the natural language (sub-)instructions are perfectly aligned in the expert demonstrations. This would be a remarkably strong assumption, compared to other formulations, where the natural language refers to an abstract, high-level goal (with multiple implicit subgoals), and where the actions are at a finer granularity than the high-level goal *and* its implicit subgoals.

Section 3, Conclusion: Missing discussion regarding the limitations of using a predefined lexicon and inability to generalize to arbitrary natural language instructions (as in the popular instruction-following tasks: ALFRED, TEACh, R*R, etc.)

Section 3, Section 4: Missing more complex examples, such as nested prepositional phrases, as alluded to in Section 3

Section 4: The ablation experiment consists of the proposed model under perfect perception conditions (ground-truth visual grounding), compared with an instance of itself without perfect perception, and shows better results under perfect perception. Why is this considered an "ablation study", and why is it an informative experiment?

**Summary Of The Paper:**

This manuscript considers the task of vision-language manipulation; the proposed approach leverages Combinatory Categorical Grammar (CCG) to parse natural language instructions into a manipulation program (task plan from a domain-specific language), consisting of functional modules.

**Summary Of The Review:**

Limited novelty: lacking insights from new methodology or optimization techniques; manuscript predominantly provides an engineering solution

The problem formulation relies on strong assumptions: regarding the definition of the DSL and regarding the alignment between natural language instructions and the observations/actions; it is not clear that the proposed approach would generalize to more complex instruction-following tasks

---

### Decision · Program_Chairs · 2023-01-20

**Decision:**

Accept: notable-top-25%

**Justification For Why Not Higher Score:**

I share a positive passion for this work, together with the reviewers. Still, the concern is only a principled engineering approach for vision-language manipulation, using a combination of programmatic functions via deterministic models and layer-weight initialization strategies via pre-trained models, not as ground-breaking as an oral paper should be.

Spotlight might be the right rack to put this draft, I am okay with putting it on poster rack as well.

**Justification For Why Not Lower Score:**

The paper presents a novel way of training a robotic manipulation model with help from a Vision Language model for localization of novel objects. It solves a problem encountered in a previous study where it seems that the model learned was unable to efficiently disentangle aspects of scene understanding from aspects of action generation. The architecture is able to efficiently learn to manipulate unseen objects or seen objects in manipulation tasks where they were not observed before. An ablation experiment shows that the action module performs nearly perfectly when the VL outputs are replaced with ground-truth segmentation.

**Metareview: Summary, Strengths And Weaknesses:**

This paper addresses the problem of solving visual-language (VL) grounding tasks by leveraging pre-trained VL models. It is motivated by the fact that conventional methods are not data-efficient in the training stage and have poor generalization to unseen objects and tasks. The study takes advantage of the stength of VL models to narrow down (mask) the relevant locations for start and end of manipulation tasks (pick A and place in B where A and B can be rich object and location entity descriptions). The results show significant improvements across several manipulation tasks in terms of performance, generalization to unseen entities and across task types while also exhibiting better data efficiency (within the constrained domain) compared with the previous work.

+ This paper is an exemplary case of harnessing general-purpose modeling of the world (scenes and their descriptions) to solve an engineering problem within a limited domain (a small set of manipulation tasks involving two objects / entities)
+ The motivation for making visual grounding tasks more generalizable by utilizing compositional program-based modules is convincing and intuitive.
+ The proposed an action module that is conditioned on the grounding map and the extended architecture from CLIPort seem effective for preventing overfitting.

Reviewers all suggested revising points to improve the final draft, please incorporate them in properly.

**Note From Pc:**

if the above contains the word "oral" or "spotlight" please see: "oral" presentation means -> notable-top-5% and "spotlight" means -> notable-top-25%. As stated in our emails, we are disassociating presentation type from AC recommendations

**Summary Of Ac-Reviewer Meeting:**

n/A